# SWIS - Shared
# Weight bIt Sparsity for Efficient Neural Network Acceleration

## ABSTRACT

Quantization is spearheading the increase in performance and efficiency of neural network computing systems making headway into commodity hardware. We present SWIS - Shared Weight bIt Sparsity, a quantization framework for efficient neural network inference acceleration delivering improved performance and storage compression through an offline weight decomposition and scheduling algorithm. SWIS can achieve up to 52% (19.8%) point accuracy improvement when quantizing MobileNet-v2 to 4 (2) bits post-training (with retraining) showing the strength of leveraging shared bit-sparsity in weights. SWIS accelerator gives up to 6X speedup and 1.8X energy improvement over state of the art bit-serial architectures.

## ACM Reference Format:

. 2020. SWIS - Shared Weight bIt Sparsity for Efficient Neural Network Acceleration. In *Proceedings of* . ACM, New York, NY, USA, 8 pages. https://doi.org/10.1145/nnnnnnn.nnnnnnn

## 1 INTRODUCTION

Creating custom silicon for a particular application historically requires a robust economic case due to the immense costs of such endeavors. Deep neural networks (DNNs) have created such a case in a span of a few short years and both training and inference accelerators are proliferating in server- and edge-class devices [7]. Many of those are doubling down on further specialization to squeeze out more efficiency, frequently through the use of quantization going as low as 4-bit or binarized precision [9]. However, only a subset of applications can take advantage of such aggressive precision reduction.

Recently, a lot of research interest has gone into hardware supporting configurable levels of quantization, for example bit-serial and decomposable arithmetic [8, 11, 13]. More recent works utilizing bit-serial arithmetic have attempted avoiding ineffectual computations resulting from zero-valued bits, however they applied it only to activations at runtime [1, 3]. Those approaches lead to limited latency improvements [3], significant hardware overheads [1, 3], no storage compression [3], or non-trivial scheduling issues [1]. Moreover, most existing bit-serial, precision-scalable architectures show benefits when quantizing from 16-bit networks [3, 8]. Recent efforts have shown that 8-bit quantization does not lose accuracy for most networks [6] and therefore, value of precision-scalable approaches needs to be shown *below* bitwidth of 8.

To address these issues, we propose SWIS - Shared Weight bIt Sparsity Scheduling, a methodology for training, compressing, and executing convolutional neural networks on bit-serial hardware that can significantly reduce the effective required bitwidth. SWIS

, ,
© 2020 Association for Computing Machinery.
ACM ISBN 978-x-xxxx-xxxx-x/YY/MM...$15.00
https://doi.org/10.1145/nnnnnnn.nnnnnnn

achieves this through configurable, non-consecutive shift values on a very fine granularity of small groups of weights. This results in both efficient hardware implementation, as well as more compressed representation. Thanks to offline profiling of weights, SWIS can achieve significant storage compression and efficient scheduling, which is not achievable in accelerators that process activations in a bit-serial manner.

The main contributions of this work are as follows.

- We show that *Shared bit sparsity* can achieve up to 3.7X neural network weight compression compared to conventional quantization approaches at similar inference accuracy.
- The proposed SWIS architecture gives up to 6X (1.8X) improvement in inference latency (energy) compared to state-of-the-art bit-serial accelerators of same size.
- We develop *filter scheduling* approaches that maximize the benefits of SWIS by optimizing distribution of shift cycles among filters on a fine granularity, giving up to 3.5p.p. improvement in accuracy over unscheduled version.

## 2 SWIS QUANTIZAION

In this section, we discuss how SWIS quantizes weights and how it can be a more effective lossless as well lossy quantization strategy.

### 2.1 What Should be Quantized?

Quantization and reduced precision have proven to be low-hanging fruits for improving the efficiency of neural network inference [8, 10, 13, 17]. When these techniques are applied, a question arises - which values should be quantized, weights or activations? Commodity hardware, like CPUs or GPUs, will often enforce symmetric quantization, with both weights and activations using the same precision, while conventional bit-serial hardware can only effectively quantize one of the two [8]. Most bit-serial work has opted for reducing the precision of activations while keeping weight precision unchanged [1, 8]. We will now argue that this approach is flawed and that reducing the precision of weights should be prioritized in such architectures.

Firstly, prior works have shown that weights can be quantized much more aggressively than activations without significant accuracy drops [10, 17]. With quantization-aware training, weight precision can be reduced to just 1 or 2 bits, and results for post-training quantization also suggest quantizing weights to lower precision is better than doing the same for activations in most cases [2]. Unlike activations, weights are not input-dependent; thus, they can be quantized offline at a much finer granularity without inducing hardware overheads. Architectures that use different precision weights and activations have opted to reduce precision more on the weight side [13], except for the aforementioned bit-serial accelerators.

Secondly, there are performance considerations. In modern DNNs, the overall number of weights will often dwarf the number of intermediate activations generated. Consider the ratio of external memory weight to activation accesses in the ResNet-18 model, shown in Figure 1, for a systolic array accelerator. For some convolutional layers, there can be two orders of magnitude more weight than activation accesses. Considering how system performance can be dominated

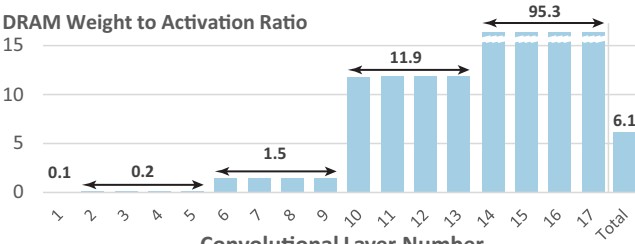

**Figure 1: Ratio of DRAM weight to activation accesses (RD+WR) in different convolutional layers of ResNet-18 in a systolic array accelerator.**

by memory accesses, reducing the precision of weights can yield much greater improvements than doing this for activations.

We will now describe SWIS - a computation scheme that can quantize weights in a much more efficient manner than traditional bit-serial approaches.

## 2.2 Shared Weight Bit-Sparsity

The multiply-accumulate (MAC) operation, which is the workhorse of deep neural networks, between an activation vector $\vec{a}$ and weight vector $\vec{w}$ can be written as:

$$\vec{a} \cdot \vec{w} = \sum_{i=0}^{M-1} a_i \times w_i \qquad (1)$$

Where $a_i$ and $w_i$ are the i-th elements of vectors $\vec{a}$ and $\vec{w}$ respectively and $M$ is the width of the multiply-accumulate. We will refer to the $M$ as a group size from now on. Each weight $w_i$ can be further decomposed to its bit-wise form:

$$w_i = Sign(w_i) \times \sum_{j=0}^{B-1} 2^j \times w_i[j] \qquad (2)$$

Where $w_i[j]$ is the j-th bit (from LSB) of weight $w_i$, and $B$ is the bitwidth of the weight. Equation 1 can now be rewritten as:

$$\vec{a} \cdot \vec{w} = \sum_{j=0}^{B-1} 2^j \sum_{i=0}^{M-1} Sign(w_i) \times a_i \times w_i[j] \qquad (3)$$

If we consider that multiplication by a single bit is a bit-wise AND operation (&), and multiplication by a power of 2 is a logical shift operation (<<), Equation 3 can be rewritten as:

$$\vec{a} \cdot \vec{w} = \sum_{j=0}^{B-1} \left( \sum_{i=0}^{M-1} Sign(w_i) \times (a_i \& w_i[j]) \right) << j \qquad (4)$$

The above formulation is used in bit-serial accelerators, although most prior works use activations in their bit-serial representation and weights in their parallel representation [3, 8]. We will now explain why the weight bit-serial formulation, as in Equation 4, can be much better.

Naive implementation of bit-serial multiplication requires going through all bits of one of the operands. However, as multiple previous works have pointed out, every bit equal to 0 will not contribute to the final result, effectively wasting computation cycles [3]. One solution is to clip all MSB and LSB positions containing zeroes and only process bits within that clipped range [3]. However, that does not eliminate zero-bits within the clipped range. For example, the above scheme applied to a value of 129, represented as an 8-bit value

(*1000_0001* in binary), results in no cycle savings, despite 75% of bits not contributing to the result.

Further, this will cause synchronization problems that are difficult to solve in highly-parallel architectures unless the above scheme is applied on a group basis [1]. However, when applied to a group of values, clipping is constrained by the worst-case number, reducing achievable benefits. Consider grouping 129 (*1000_0001* in binary) with 8 (*0001_0000*). The former will require processing all 8-bit positions, while the latter only requires a single one. Overall, over 80% of computation would effectively be wasted. While more sophisticated techniques of removing all activation zero bit computations have been proposed, they suffer from the above synchronization issue and significant hardware overheads. [1]. While training optimizations for such architectures have recently been proposed, they do not fully solve the scheduling issues [16].

What limits the efficacy of the methods described above is that they are attempting what is effectively "lossless compression" of computation, requiring representation of exact values. We argue that through careful pre-processing, a much more hardware-friendly "lossy compression" can be achieved without significantly reducing inference accuracy, as we will show in Section 5.1. However, pre-processing implies that it can only be applied to weights and not activations, which are input dependent. This insight, together with the reasons outlined in Section 2.1 justify our "reverse" weight bit-serial formulation in Equation 4. Furthermore, these existing approaches quantize using consecutive bit positions (usually truncating the LSBs). Next we show SWIS approach to leverage the sparsity in bit representations of weights.

Let us assume we constrain a group of weights to only use a specific subset of *active* bit positions, while all the other *inactive* positions are assumed to be 0. We can define a supporting vector $\vec{s}$:

$$\vec{s} = (s_0, s_2, ..., s_{N-1}) : s_i \in \langle 0, B \rangle \qquad (5)$$

We can then rewrite Equation 2 as:

$$w_i = Sign(w_i) \times \sum_{j=0}^{N-1} 2^{s_j} \times m_i[j] \qquad (6)$$

Where $m_i$ is a *mask* bit indicating whether weight $w_i$ has an active bit in position $s_j$. After combining Equations 4 and 6 we arrive at the shared weight bit sparsity formulation, the foundation of the SWIS methodology:

$$\vec{a} \cdot \vec{w} = \sum_{j=0}^{N-1} \left( \sum_{i=0}^{M-1} Sign(w_i) \times (a_i \& m_i[j]) \right) << s_j \qquad (7)$$

The stark similarity between Equations 4 and 7 means that SWIS is fully compatible with bit-serial MAC processing elements (PEs). There are three crucial differences between bit-serial and SWIS processing. First is the change in the outer loop bound from $B$ (weight bitwidth) to $N$ (size of the support vector). Second is the sparse (non-consecutive) nature of the supporting vector - most prior bit-serial architectures either constrained themselves to consecutive shift ranges [8], or ran into non-trivial scheduling problems when attempting to exploit bit-sparsity in dynamic activations [1]. SWIS does not have this problem as long as the number of active bits, henceforth referred to as *shifts*, is the same for all computations scheduled at the same time.

The third difference is the flexibility to select shifts on the granularity of an individual group. Traditional bit-serial approaches constrain

themselves to per-layer profiling of consecutive shifts, which, as we will show in Section 2.3, can be overly restrictive. We refer to this approach as *layer-wise static quantization*. Through a careful selecting and scheduling approach, described in Section 4.1 and 4.3, SWIS can ensure that $N << B$, without sacrificing inference accuracy.

Recent works have shown that using consecutive shifts with a fine-granularity can also yield acceptable accuracy for certain datasets and networks [15]. SWIS can support consecutive shifts without any additional overheads while taking advantage of the higher weight compression ratio enabled by it, since only a single *shift offset* needs to be stored per group of weights, instead of individual sparse shift values. We refer to this configuration as SWIS-Consecutive, or SWIS-C for short. The important distinction between SWIS-C and typical quantization approaches is that the *offset* being used can be set on a very fine granularity of a group of weights, instead of a per-kernel or per-layer basis, hence allowing more aggressive quantization without sacrificing accuracy.

## 2.3 Granularity of Weight Quantization

We discuss the relative accuracy of three quantization approaches in this section, namely layer-wise static quantization, SWIS-C, and SWIS. To establish the superiority of both SWIS methods, we will first discuss their approximation ability, which can be reflected by the probability of losslessly quantizing an 8-bit integer $A$ into $\bar{A}$ using a given number of shifts $N$. The 8-bit number is assumed to be randomly generated so that each bit will have a 50% probability of being one.

First, for SWIS, as the bit selection is sparse, the quantization is lossless if the number of bits being 1 in $A$ is smaller or equal to $N$. The probability of lossless quantization for SWIS given $N$ can be formulated using cumulative binomial distribution:

$$P_{SWIS}(A == \bar{A}) = \sum_{n=0}^{N} \binom{8}{n} \cdot 0.5^8 \tag{8}$$

Second, for SWIS-C, the probability formulation is more complicated due to the constraint of consecutive shift values. For each $N$, it can be calculated based on the probability of SWIS, multiplied by the fraction of total bit permutations that can be losslessly quantized. The probability of lossless quantization of SWIS-C for given $N$ can therefore be formulated by:

$$P_{SWIS-C}(A == \bar{A}) = \sum_{n=0}^{N} \binom{8}{n} \cdot 0.5^8 \cdot \frac{\binom{N}{n}(9-N) - (8-N)\binom{N-1}{n}}{\binom{8}{n}} \tag{9}$$

Last, for layer-wise static quantization, the bit selection is fixed for entire layer, therefore the probability of lossless quantization of an individual 8-bit value is:

$$P_{layer-wise}(A == \bar{A}) = \sum_{n=0}^{N} \binom{8}{n} \cdot 0.5^8 \cdot \frac{\binom{N}{n}}{\binom{8}{n}} \tag{10}$$

Figure 2 shows the computed probability of lossless quantization for all three approaches at every $N$. The results are expected, SWIS outperforms the other two by a large margin in most cases due to its bit sparsity, while SWIS-C also outperforms layer-wise quantization noticeably, since it allows a finer granularity.

The relative accuracy of lossless quantization also holds for lossy quantization. We use root mean square error (RMSE), instead of probability, to compare the above three methods. Table 1 shows quantization RMSE against original weights for a typical layer of

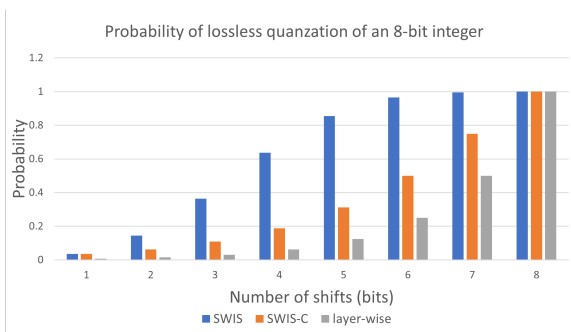

**Figure 2: Probability of lossless quantization of a random 8-bit integer using layer-wise static quantization, SWIS-C and SWIS.**

8-bit ResNet-18 [4] and MobileNet-v2 for a different number of shift values. To measure the upper bound of accuracy, we set the group size to 1, meaning that shift values are chosen for each weight individually. We will show in Section 4.2 how using a larger group size affects overall accuracy. Both networks show a similar trend, and the huge RMSE of static layer-wise quantization (implemented using LSB truncation) suggests that it does not work well for lower bit widths. SWIS outperforms SWIS-C in all cases, and the gap is large for the combination of a hard-to-quantize network (MobileNet-v2) and a small number of shift values. However, SWIS-C shows overall good performance and can be considered an alternative for some networks, with a better weight compression than SWISS, as discussed in Section 3.3.

**Table 1: RMSE of three weight quantization methods for a typical layers of 8-bit ResNet-18 and MobileNet-v2, assuming the group size of one.**

| # shifts | SWIS | SWIS-C | layer-wise truncation |
|---|---|---|---|
| ResNet-18 first convolution layer | | | |
| 5 shifts | 0.0013 | 0.0020 | 0.0168 |
| 4 shifts | 0.0019 | 0.0037 | 0.0314 |
| 3 shifts | 0.0038 | 0.0070 | 0.0556 |
| 2 shifts | 0.0094 | 0.0146 | 0.0895 |
| MobileNet-v2 first point-wise convolution layer | | | |
| 5 shifts | 0.0007 | 0.0011 | 0.0158 |
| 4 shifts | 0.0009 | 0.0043 | 0.0227 |
| 3 shifts | 0.0031 | 0.0098 | 0.0394 |
| 2 shifts | 0.0104 | 0.0186 | 0.0774 |

## 3 ARCHITECTURE

We architect SWIS as a bit-serial processed systolic array with each processing element (PE) and dataflow optimized to leverage SWIS quantization.

## 3.1 SWIS PE

The conventional processing element (PE) implementation of Equation 7 would consist of N (group size) parallel bitwise AND operations (masking), conditional sign inversion, an adder tree for summing masked activations, a barrel-shifter for power-of-2 multiplication

and a serial accumulator, similar to the one proposed in [8]. It computes one of the operands one bit at a time. We refer to this style of bit-serial PE as a *single-shift* PE. While inverting the order of addition and multiplication results in certain gains in efficiency, bit-serial processing by itself does not provide higher throughput per area or energy efficiency compared to conventional fixed-point when processing all of the bits. Only by aggressively reducing the number of bits (shifts) being used and maximizing the PE group size, performance improvements over fixed-point can be achieved. While such improvements are trivial when 16-bit fixed-point precision is used as a baseline, they are much harder when the baseline is reduced to 8-bits, the de-facto standard precision in quantized networks nowadays. [8].

To quantify the possible benefits of using bit-serial computation, we have designed the 8-bit fixed-point, and a single-shift bit-serial PEs with different group sizes (2-16) using Verilog RTL and synthesized them using a commercial 28nm TSMC library and Cadence Genus synthesis tool. Since we intended to use them in a systolic array style accelerator, all PEs include activation and weight buffers. We then compared their area, energy per MAC, and throughput per area for different number of shifts used in the bit-serial version (2/4/6). Results, normalized to the fixed-point PE are shown in Figure 3. The single-shift PE only comes out ahead in terms of energy and throughput per area when fewer than 4 shifts are used. When using conventional quantization approaches, this level of precision reduction might not be tolerable, as we will show in Section 5.1 SWIS, with its ability to implement sparse quantization on a much finer granularity, can reduce the number of shifts required much more aggressively than those approaches.

However, even with SWIS, improving the performance requires using PEs with large group sizes, as shown in Figure 3. Below a group size of 8, performance improvements, even with a low number of shifts used, are modest at best. This limited improvement is due to overheads which cannot be reduced compared to fixed-point PEs. As we will show in Section 4.2, a larger group size makes it much harder or even impossible for SWIS to recover accuracy. Therefore, a way to improve hardware efficiency is needed. To better amortize the fixed costs mentioned above, we propose to process multiple bits (shifts) simultaneously. By computing, for example, two shifts at the same time, performance break-even points compared to fixed-point can be improved.

We show the performance comparison of this *double-shift* PE in Figure 3, for the same group sizes and number of shifts being used as the single-shift one. It has a lower normalized energy per MAC and throughput per area than a single-shift one with double the group size. This means we can effectively halve the group size while improving both performance and inference accuracy. For that reason, we opt to use double-shift PEs in our SWIS accelerator architecture, as shown in Figure 4. However, this double-shift processing comes at an increased rigidity in terms of the number of shifts used. Using an odd number of shifts would result in underutilization of the available compute - going from four to three shifts would therefore not improve inference latency. However, SWIS allows us to assign the number of shifts on a sub-layer granularity, meaning that *effective* number of shifts is not constrained to even numbers. For example, if half of the kernels in a given layer use 2 shifts, and the other half 4 shifts, the effective, layer-wise number of shifts is 3. See Section 4.3 for network accuracy when using a scheduled odd effective number of shifts on the *double-shift* configuration.

## 3.2   SWIS Systolic Array and Dataflow

We use systolic array as a baseline architecture, shown in Figure 4, due to simple scheduling, low complexity processing element architecture, and low bandwidth requirements when processing convolutional layers [7]. We assume the same structure, consisting of the systolic array itself, together with activation, weight, and output buffers, as described in [12]. That being said, SWIS is not inherently tied to a particular implementation and could be used in any accelerator that can support bit-serial processing.

Compared to conventional systolic array, where each element consists of a single multiplier and accumulator, SWIS systolic array uses group-wise PEs, where multiple MAC operations are executed in parallel on a vector of activations and a corresponding vector of weights, one shift at a time. For simplicity, we assume that all such vectors are depth-wise - all activations and weights have the same x and y positions but correspond to different input channels. We also assume that those vectors are packed in memory, and on-chip buffers have interfaces scaled by a factor equal to the group size. Those assumptions are easy to fulfill for commonly used convolutional layers where the number of input channels is a power of 2. For depthwise-separable convolutions, such as those used by MobileNet, we underutilize the PEs in the systolic array, for the simplicity of scheduling. We plan on exploring a more efficient implementation of such layers in future work.

In terms of scheduling, we use the output stationary dataflow (OS), as it has been shown to provide the best performance and minimal number of memory accesses in most cases [12]. There are several ways bit-serial computation can be scheduled in a systolic array. The most naive would be to perform a full computational pass for each shift. While straightforward to implement in the OS dataflow, it would also increase the number of on-chip memory accesses roughly proportional to the number of shifts being used. Another alternative is to send all shift masks to the PE at the same time and execute each operation in multiple cycles. Unfortunately, this would require scaling both the weight buffer interface and PE weight buffers to support the worst case, 8 shifts, drastically increasing their area. Instead, we opt for a "staggered" approach, where weights (shifts) flow through the array normally, but each activation is fed in repeatedly over multiple cycles, equal to the number of shifts being used. Such an approach requires minimal control and buffering overhead, without over-provisioning the PE buffers or increasing the number of activation buffer accesses. For SWIS-C, we assume that a shift (offset) is fetched only once, and incremented outside of the array, incurring negligible area overheads.

## 3.3   SWIS Compression

The performance of a computing system cannot be evaluated without considering the impact of memory. Increasingly, memory bandwidth and access energy have a dominant impact on overall latency, and energy [5]. Approaches that rely solely on point improvements to arithmetic efficiency will quickly fall victim to diminishing returns. One of the main advantages of SWIS is the weight storage compression it offers. Assuming 8-bit underlying precision, for each group of weights, we only need to store their signs (one bit per weight), shift values (3 bits per group, per shift), and shift masks (1 bit per weight, per shift).

The resulting weight compression ratios for different number of shifts and group sizes are shown in Figure 5. We compare our compression scheme of 8-bit weights to the one used by DPRed [3], profiled across one example convolution layer, for different groups

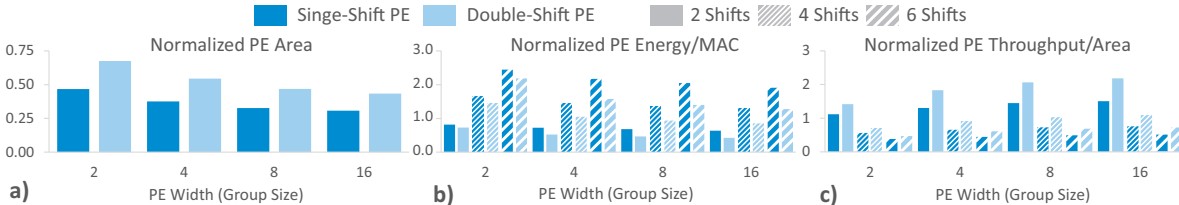

**Figure 3: Single- and double-shift 8-bit SWIS PE area (a), per-MAC energy (b) and throughput/area (c) for different PE widths, normalized to a conventional fixed-point PE with the same group size.**

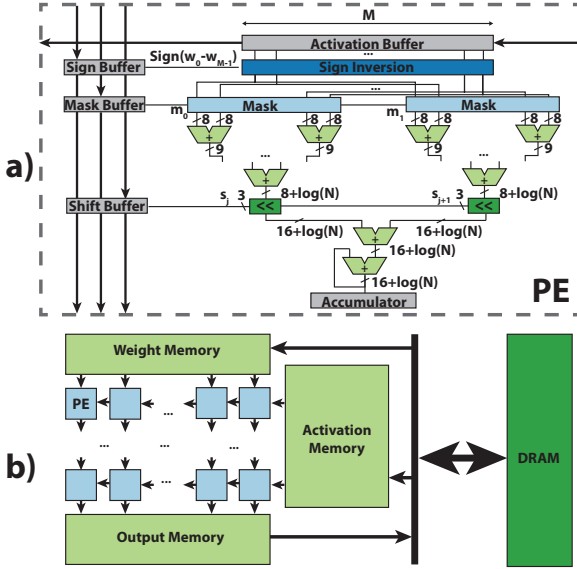

**Figure 4: N-wide double shift bit-serial MAC unit (a) and systolic array accelerator (b) used by the SWIS methodology.**

sizes. DPRed stores weights using per-group bitwidth, determined by the highest active bit position in a given group. We also show compression ratios for SWIS-C, which only needs to store one shift value per group.

While it is important to note that unlike SWIS, DPRed compression is lossless (retains all information), it is also too restrictive, at least at 8-bit precision, to deliver any significant storage savings. Meanwhile, SWIS and SWIS-C can deliver close to 3.7X reduction in weight storage when large groups are used with an aggressive reduction in the number of shifts. For a group size of 4, which we use in our architecture, compression varies between 1.1X and 2.9X, and 1.5X and 2.9X for SWIS and SWIS-C respectively. Accuracy-performance trade-offs between the number of shifts and group sizes are explored in Section 5.

## 4 SWIS SCHEDULING & GROUPING

### 4.1 SWIS Shift Selection

The shift selection process for SWIS consists of selecting the optimal shift values $s_j$ for each group and generating the bitmasks $m_i$ for individual weights to minimize the quantization error for the given number of shifts. As the total number of possible combinations of selecting $N$ shift values out of 8 is manageable, we use an enumeration

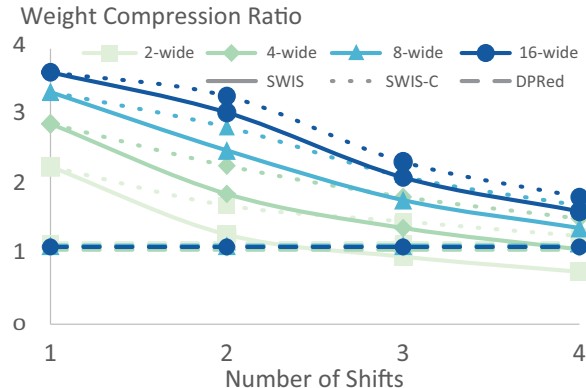

**Figure 5: Weight storage compression ratio for different number of shifts and PE sizes, for SWIS, SWIS Consecutive, and DPRed.**

algorithm. For each group, we quantize the weights using all possible shift value combinations and select the combination with the least RMSE over the entire group. For each shift value combination, the corresponding values for all possible bitmasks are generated, and each weight is quantized to the nearest value (bitmask). This enumeration algorithm ensures that the optimal shift values and bitmasks are selected for every group and every weight to minimize the RMSE error.

### 4.2 SWIS Grouping

The previous analysis of different quantization granularities assumes that the group size is one, but that does not result in efficient hardware implementation or storage compression. However, increasing the group size will increase the quantization error and impact network accuracy as the shift values for the entire group of weights need to be shared. Figure 6 shows the inference accuracy of ResNet-18 on ImageNet, with different group sizes and number of shift values. As expected, inference accuracy drops as group size increases, but the exact amount differs significantly for different number of shift values. SWIS performs better than SWIS-C when the number of shift values is small, but their performance converges when the number of shift values increases, which verifies the analysis in section 2.3. For a group size of 4, which tends to be a good accuracy/efficiency trade-off point, we need 3 shifts to maintain a similar performance of 8-bit baseline. In the next section, we will discuss how to obtain even finer granularity of the number of shits being used.

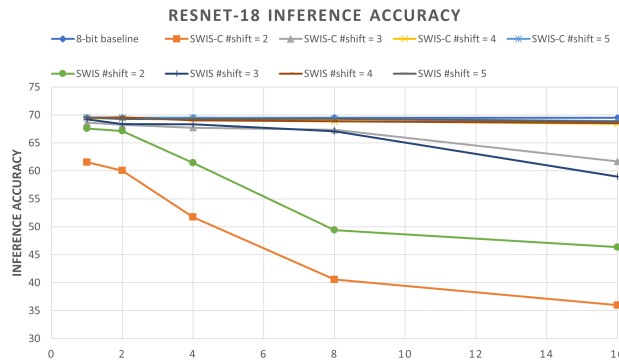

**Figure 6: ResNet-18 top-1 inference accuracy for different group sizes and number of shift values.**

## 4.3 SWIS Scheduling

Within a layer, not all filters are equally sensitive to the loss of precision. SWIS scheduling takes advantage of this to decrease the RMSE for a given layer compared to the RMSE achieved by naively quantizing the entire layer to the same number of shifts. We do this by increasing the number of shifts for some filters while decreasing it for others to keep the total number of shifts constant for the layer. This scheduling approach's main benefit is that it allows us to choose an average quantization level that would not be possible without filter scheduling. For instance, it allows the *double-shift* architecture to use a target number of shifts that is not an even number without under-utilizing the hardware.

The SWIS scheduling heuristic starts by placing all filters at a number of shifts higher than the target value. We then calculate the RMSE cost of decreasing the number of shifts used to quantize each filter by one shift. The filters are then sorted based on this cost, and the lowest cost *n* filters are moved down to the next lowest shift. The new cost for the filters which changed their number of shift values are then recomputed, and the filter costs are sorted again to find the *n* lowest cost filters. This process is repeated until the average number of shifts in the layer is equal to the target number of shifts. At this point, the filters are sorted based on their number of shifts.

The above method does not guarantee that all filters scheduled simultaneously on the systolic array have the same number of shifts, a restriction that is necessary to ensure simple scheduling and the absence of synchronization issues. To enforce such behavior, the second part of the algorithm assigns the number of shifts to each group of filters that are scheduled simultaneously, based on previous ordering. We first enumerate the possible per-filter-group number of shift assignment sequences that are nondecreasing and guarantee the desired overall average number of shifts per layer. For each sequence, we compute the RMSE and select the combination with the lowest RMSE.

Accuracy using SWIS scheduling for *single-shift* improved as shown in Table 2.

## 5 EVALUATION & RESULTS

All PE area, power, latency numbers are derived from synthesis results in a commercial 28nm library with Cadence Genus tool. We used SCALE-Sim, a systolic array simulator, to obtain cycle-accurate execution traces [12]. As a baseline, we used an 8x8 bit-serial systolic

**Table 2: ResNet-18 top-1 accuracy with SWIS scheduling for single- and double-shift PEs, compared to a single-shift PE accuracy with no scheduling for different systolic array (SA) sizes. PE group size is 4.**

|  | 2 Shift % Accuracy | | | 2.5 Shift % Accuracy | | |
|---|---|---|---|---|---|---|
| SA | Single | Double | None | Single | Double | None |
| 8 | 65.07 | 64.94 | 61.42 | 67.95 | 67.1 | N/A |
| 16 | 64.21 | 61.42 | 61.42 | 67.44 | 67.18 | N/A |
|  | 3 Shift % Accuracy | | | 4 Shift % Accuracy | | |
| 8 | 68.77 | 68.47 | 68.31 | 69.48 | 69.32 | 69.05 |
| 16 | 68.79 | 67.84 | 68.31 | 69.38 | 69.05 | 69.05 |

array with 64KB activation and weight buffers, and 16KB output buffer. The PE group size has been set to 4, as it provides a good balance between performance and accuracy. We compare the following versions of SWIS:

- SWIS-DS - *double-shift* SWIS.
- SWIS-SS - *single-shift* SWIS.
- SWIS-C-DS - *double-shift* SWIS consecutive.
- SWIS-C-SS - *single-shift* SWIS consecutive.

As a baseline, we use a systolic array with conventional (*single-shift*) bit-serial PEs using per-layer activation truncation. Computation is done in the same way as [8], however the accelerator organization is different. We also compare to the same architecture, but using weight truncation. Finally, we compare SWIS to BitFusion, a systolic array using decomposable arithmetic [13]. The area and energy numbers have been scaled appropriately to 28nm, whenever necessary. We evaluate BitFusion using 4-bit weights and 8-bit activations, as the architecture is constrained to power-of-2 precision. All configurations have the same amount of on-chip memory. All comparison points use the same size of the systolic array (8x8) as it allows us to isolate the benefits coming from each scheme. We evaluate the performance only on convolutional layers of tested networks, as they dominate overall performance and latency. We leave SWIS optimizations targeting fully-connected layers for future work.

For network accuracy evaluation, we use Pytorch as the framework and implement all custom quantization functions using Pytorch's built-in functions. Table 3 shows the networks and datasets we used as benchmark and their baseline accuracy. We select ResNet-18 and MobileNet-v2 on ImageNet 2012 and VGG-16[14] on CIFAR100 to evaluate the results. For MobileNet-v2, the floating point weights are downloaded from Pytorch's model zoo and then retrained for 10 epochs with 8-bit quantization to generate the 8-bit baseline weights, as MobileNet-v2 performs poorly on post-training INT8 quantization. For ResNet-18, the 8-bit baseline is the layer-wise static INT8 quantization of pytorch's pretrained floating point weight. For VGG-16, the network structure is adjusted slightly to fit CIFAR-100 dataset and trained from scratch for 100 epochs to obtain the floating point accuracy. The INT8 baseline is the layer-wise static quantization of floating point network. For quantization-aware retraining, all baseline results are trained for 10 epochs with learning rate decay. Some SWIS variants also fine-tune based on scheduling algorithm's output to enable odd number of shifts (for DS) and half shifts. All activations are also quantized to 8 bits unless specified.

For SWIS weight quanization, we use the method introduced in Section 4.1. To simulate the activation quantization used in [3, 8], we implement a layer-wise LSB truncation algorithm on all activations,

**Table 3: Networks and datasets used for benchmark, with their top-1 accuracy**

| Network | Dataset | FP32 Accuracy | INT8 Accuracy |
|---|---|---|---|
| ResNet-18 | ImageNet | 69.6% | 69.5% |
| MobileNet-v2 | ImageNet | 71.9% | 70.1% |
| VGG-16 | CIFAR100 | 64.8% | 64.8% |

where the last $8-N$ bits are truncated and $N$ is the number of shifts allowed.

## 5.1 Network Accuracy Evaluation

*5.1.1 Post-training Quantization.* In this section we compare the accuracy of the 4 SWIS configurations to layer-wise activation truncation (similar to the approach used in [8]) and layer-wise weight truncation + clipping, which is a standard baseline method for weight quantization. Table 4 shows the post training quantization accuracy for all quantization configurations on the three networks shown in table 3. All SWIS/SWIS-C results are after scheduling. All four SWIS configurations outperform weight and activation truncation by a large margin in all cases. In general, SWIS outperforms SWIS-C and SS outperforms DS slightly due to better scheduling flexibility. In most cases, the accuracy difference between DS and SS is small, and DS should be preferred due to its better hardware efficiency. The accuracy difference between SWIS and SWIS-C depends on networks, the gap is relatively small for more redundant networks like VGG-16 on CIFAR100 while it is large for MobileNet-v2, where SWIS shows advantage of its bit sparsity quantization. Post-training activation quantization (as in [8]) below 8 bits has unusably low accuracy. Even for weight quantization, for example at 4 bits (or shifts), SWIS has 9.3%, 52%, 1.5% higher accuracy than conventional quantization for Resnet-18, MobileNet-v2 and VGG-16 respectively.

*5.1.2 Quantization-aware Retraining.* Though our focus is to show energy/latency benefits of SWIS without needing to retrain, retraining can reduce the number of shift values needed further by 1-3. Retraining is especially helpful for the MobileNet-v2 case, as it needs larger number of shifts to maintain accuracy for post-training quantization compared to other networks. Table 5 shows the retraining results for the three networks, all SWIS configurations still outperform weight truncation in all cases (5%, 19.8%,4.5% point accuracy gain over conventional quantization at 2-shifts for the three networks). SWIS at 2 shifts in all its variants is *far superior* in accuracy compared to conventional quantization at 3 shifts.

## 5.2 Performance Comparison

Performance results, in terms of frames per Joule (F/J) and frames per second (F/s) for each evaluated configuration are listed in Table 6. Performance for each SWIS configuration is evaluated at 2 accuracy points, with corresponding activation- and weight-truncation results, as well as BitFusion 4x8 where applicable. First, we show that SWIS-SS can be between 1.75x and 3x faster than activation-truncation bit-serial. For SWIS-DS that speedup ranges from 2.8X to 6X. SWIS can also improve energy efficiency by 1.04-1.7X and 1.1-1.8X for SWIS-SS and SWIS-DS respectively, due to weight compression and more efficient computation. When using the same number of shifts, SWIS-C has higher energy efficiency than SWIS, but that benefit is often offset when additional shifts are required to maintain iso-accuracy with it.

**Table 4: Post-training quantization top-1 accuracy of the three networks, using different algorithm and hardware setups, Wgt. and Act. means weight truncation and activation truncation. Results for weight and activation truncation with 6 and 7 shifts are included for reference.**

| N_shift | SWIS SS | SWIS DS | SWIS-C SS | SWIS-C DS | Trunc. Wgt. | Trunc. Act. |
|---|---|---|---|---|---|---|
| | | | Resnet-18 ImageNet | | | |
| 2 | 65.1 | 64.9 | 62.4 | 59.1 | 3.6 | 0.1 |
| 2.5 | 68.0 | 67.1 | 65.9 | 65.6 | N/A | N/A |
| 3 | 68.8 | 68.5 | 68.1 | 67.4 | 30.8 | 0.1 |
| 4 | 69.5 | 69.3 | 69.5 | 69.4 | 60.2 | 45.9 |
| 6 | / | / | / | / | 69.2 | 66.7 |
| 7 | / | / | / | / | 69.5 | 69.1 |
| | | | MobileNet-v2 ImageNet | | | |
| 3 | 30.7 | 5.3 | 6.9 | 3.7 | 0.6 | 0.1 |
| 3.5 | 52.9 | 44.6 | 34.0 | 50.8 | N/A | N/A |
| 4 | 65.2 | 65.2 | 57.7 | 57.7 | 13.2 | 0.3 |
| 5 | 69.1 | 67.2 | 67.6 | 64.5 | 60.6 | 25.8 |
| 6 | / | / | / | / | 68.0 | 60.3 |
| 7 | / | / | / | / | 70.1 | 68.1 |
| | | | VGG-16 CIFAR100 | | | |
| 2 | 57.8 | 57.8 | 57.5 | 57.5 | 31.1 | 1.0 |
| 2.5 | 62.1 | 60.6 | 60.8 | 60.3 | N/A | N/A |
| 3 | 64.0 | 62.3 | 62.7 | 62.2 | 60.5 | 3.6 |
| 4 | 64.7 | 64.7 | 64.6 | 64.6 | 63.2 | 24.7 |
| 6 | / | / | / | / | 64.7 | 62.8 |
| 7 | / | / | / | / | 64.9 | 64.1 |

**Table 5: Retraining top-1 accuracy of the three networks, using different algorithm and network setups**

| N_shift | SWIS SS | SWIS DS | SWIS-C SS | SWIS-C DS | Trunc. Wgt. |
|---|---|---|---|---|---|
| | | | Resnet-18 ImageNet | | |
| 2 | 68.3 | 68.3 | 68.1 | 68.1 | 63.3 |
| 3 | 69.1 | 68.7 | 68.4 | 68.3 | 66.3 |
| | | | MobileNet-v2 ImageNet | | |
| 2 | 67.4 | 67.4 | 65.5 | 65.5 | 47.6 |
| 2.5 | 68.0 | 67.8 | 66.9 | 66.0 | N/A |
| 3 | 69.3 | 68.5 | 69.0 | 67.2 | 65.8 |
| | | | VGG-16 CIFAR100 | | |
| 2 | 64.1 | 64.1 | 64 | 64 | 59.6 |

Even when comparing to more efficient bit-serial with weight truncation, SWIS offers offers up to 1.6X and 2.5X speedup for SWIS-SS and SWIS-DS respectively, with up to 1.55X reduction in energy across all SWIS configurations. Compared with BitFusion, at the same accuracy, SWIS can have up to 2X lower latency and up to 1.9X lower energy consumption. This is thanks to the SWIS's ability to reduce the number of bits used much more aggressively than conventional approaches can, improving both storage compression and computation energy efficiency.

**Table 6: Energy (Frames/J) and latency (Frames/s) comparison between different SWIS configurations, bit-serial with activation and weight truncation and BitFusion, at different accuracy points for different network models and datasets.**

| Architecture | SWIS | | | | | | SWIS-C | | | | | | Trunc | | | | | | Bit Fusion 4x8 | | |
|---|---|---|---|---|---|---|---|---|---|---|---|---|---|---|---|---|---|---|---|---|---|
| | SS | | | DS | | | SS | | | DS | | | Act | | | Wgt | | | | | |
| Area [mm2] | 0.54 | | | 0.55 | | | 0.54 | | | 0.55 | | | 0.54 | | | 0.54 | | | 0.57 | | |
| Network | ResNet-18 ImageNet | | | | | | | | | | | | | | | | | | | | |
| Accuracy | #S | F/J | F/s | #S | F/J | F/s | #S | F/J | F/s | #S | F/J | F/s | #S | F/J | F/s | #S | F/J | F/s | #S | F/J | F/s |
| >69.1% | 4 | 267.7 | 21.4 | 4 | 292.5 | 42.9 | 4 | 326.3 | 21.4 | 4 | 353.6 | 42.9 | 7 | 215.8 | 12.2 | 6 | 230.7 | 14.3 | N/A | N/A | N/A |
| >60.2% | 2 | 390.8 | 42.9 | 2 | 416.5 | 85.7 | 2 | 410.6 | 42.9 | 2.5 | 404.8 | 68.6 | 6 | 230.7 | 14.3 | 4 | 267.7 | 21.4 | 4 | 218.9 | 42.9 |
| Network | MobileNet V2 ImageNet | | | | | | | | | | | | | | | | | | | | |
| Accuracy | #S | F/J | F/s | #S | F/J | F/s | #S | F/J | F/s | #S | F/J | F/s | #S | F/J | F/s | #S | F/J | F/s | #S | F/J | F/s |
| >68.0% | 5 | 475.6 | 4.0 | 5 | 490.0 | 8.0 | N/A | N/A | N/A | N/A | N/A | N/A | 7 | 456.1 | 2.9 | 6 | 466.1 | 3.3 | N/A | N/A | N/A |
| >60.3% | 4 | 498.9 | 5.0 | 4 | 511.6 | 10.0 | 5 | 496.3 | 4.0 | 5 | 512.0 | 8.0 | 6 | 466.1 | 3.3 | 5 | 476.6 | 4.0 | N/A | N/A | N/A |
| Network | VGG-16 CIFAR-100 | | | | | | | | | | | | | | | | | | | | |
| Accuracy | #S | F/J | F/s | #S | F/J | F/s | #S | F/J | F/s | #S | F/J | F/s | #S | F/J | F/s | #S | F/J | F/s | #S | F/J | F/s |
| >64.1% | 4 | 625.4 | 93.5 | 4 | 626.5 | 187.1 | 4 | 815.1 | 93.5 | 4 | 843.5 | 187.1 | 7 | 553.0 | 53.4 | 6 | 569.5 | 62.4 | N/A | N/A | N/A |
| >62.1% | 2.5 | 878.2 | 149.7 | 3 | 788.5 | 249.4 | 3 | 942.1 | 124.7 | 3 | 980.3 | 299.3 | 6 | 569.5 | 62.4 | 4 | 605.6 | 93.5 | 4 | 799.8 | 187.1 |

## 6 CONCLUSION

In this work, we propose SWIS, a framework for neural network quantization for efficient inference on edge devices. We show conventional bit-serial designs do not fully utilize their flexibility as most of them only apply to activations. We utilize the bit level sparsity inherent in weights to quantize them beyond the conventional "prefix" or "suffix" style truncation. For example, SWIS quantization can achieve MobileNet-v2 accuracy within 1% of INT8 with 5 effective bit quantization *without* any retraining and 3 bits with retraining. For bit-serial architectures, SWIS compresses weights and improves latency and energy by as much as 6X and 1.8X, respectively, without loss of accuracy. Based on SWIS, we further purpose SWIS-C and double-shift SWIS (SWIS-DS), one for better weight compression and the other for better hardware efficiency. Further, we develop a filter scheduling algorithm, to allow for fine-grained tradeoff between accuracy and energy/latency. Our ongoing work includes design space exploration of SWIS systolic array architectures as well as approaches for efficient SWIS execution of fully connected layers.

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
