# OpenReview forum: "SWIS - Shared Weight bIt Sparsity for Efficient Neural Network Acceleration"
_tinyml.org/tinyML/2021/Research_Symposium — tinyML 2021 Regular_

### Official Review · AnonReviewer1 · 2021-01-28

**Overall Merit Score:** 3

**Brief Summary:**

* Propose a bit-sparse scheme that can be exploited in bit serial PE's
* Idea is the keep the bit sparsity aligned in the weights of neighboring processing elements. This allows them to speed up in cycle time
* Training algorithm proposed + benchmarked on Mobilenet and resnet
* HW architecture proposed
* energy/area/power estimated and compared to bitfusion

**Detailed Comments:**

* Explanation of SWIS-C not fully clear:
   - What does consecutive means here. And how many consecutive elements are considered?
   - Is the overhead of storing / fetching / applying the shift bits (per group) also taken into account?
* comparison not fully fair to other techniques
   - would have been good to also benchmark other techniques assuming a weight-reuse dataflow (where the same weight is reused across many PE's, and hence the same shift pattern can be reused, even without SWIS-C...)  (Basically, dataflow matters in this study, and this is not assessed).
* Some figures are confusing:
   - Not clear what fig 3 is normalized to? Why is it normalized?
   - Is the PE array 1 or 2D. In case 2-D (fig 4), what does 16-wide means: 4x4 or 16x..?


**Paper Strengths:**

* nice cross-layer optimization
* interesting idea
* rather extensively benchmarked

**Paper Weaknesses:**

* Explanation of SWIS-C not sufficiently clear
* Comparison not fully fair to other SotA techniques
* Some figures are confusing

**Poster (If Paper Is Rejected):**

1: Yes, ok for poster sesion to nurture work

**Reviewer Confidence:**

4: The reviewer is confident but not absolutely certain that the evaluation is correct

---

### Official Review · AnonReviewer2 · 2021-01-29

**Overall Merit Score:** 3

**Brief Summary:**

The authors propose a method for weight compression by restricting the number of 'active' bits in a weight (those allowed to be '1') to a number smaller than the uncompressed bit width.  They show a hardware architecture to implement and exploit the compression method.  They also explore the implications on overall model accuracy of several versions of the compression scheme and the tradeoffs between hardware performance, weight compression, and model accuracy.

**Detailed Comments:**

### Overall
The novelty in the compression scheme is minor, but the combination with the hardware simulations is interesting.  The advantages are modest and vary by NN architecture.  It is not clear that the advantages justify the limitations, but the information could be useful to other researchers, particularly if the authors can open-source their work (at least the modeling/accuracy evaluation scripts, but the verilog would be helpful also).

### Details
* (abstract) Percentage points vs percent?  Also 52% and 19.8% accuracy improvement are relative to what?

* Is it assumed that the activations are unsigned?  This would be appropriate for ReLU, but would preclude some activation functions, most notably leaky ReLU and tanh.

* Typo in eqn 5; in "s_0, s_2, ...", s_1 is skipped.

* Section 2.3, the assumption that all bits have 50% probability is not valid (at least in this reviewer's experience). Most weights are clustered at the low end of the range, so the probability for the higher-order bit will be less than 50%.  It would be useful here to actually measure the probabilities on some trained 8b model.

* Figure 2 shows the probability of losslessly quantizing a single weight, which is not very informative, given that the method relies on sharing the support vectors.  For the analysis in (8)-(10) to be useful, the probability of correctly quantizing an entire layer/group should be calculated and added to Fig. 8 for a couple of representative group sizes.  I expect that it will be very small. If so, then what is the value in have a single weight quantized  losslessly?

* Typo, end of section 2.  SWISS -> SWIS.

* Table 1 - This table should go up to much larger group sizes, since group size of 1 provides no benefit. Also, are the values expressed as a fraction of full-scale?  Please clarify.

* Section 3.2 - Can you provide some insight as to why computing two shifts in parallel is more efficient?

* Section 3.2 - Does the staggered approach the authors describe require input activations be fetched N times for N shifts?

* Figure 3.: It looks like this is showing that even for groups of 16 there is almost no improvement in energy/MAC or throughput/area for more than 2 shifts.  The text should acknowledge this.

* Typo, section 4.2 first paragraph.  "number of shits" => "number of shifts".

* Given that group size of 4 and 3 shifts seems to be the sweet spot for this algorithm, Figure 3 should also show the area/energy/throughput results for a shift of 3.  As it is, the reader has to interpolate between 2 and 4 shifts, which leaves it unclear whether there is any benefit at all.

* Is the group size and number of shifts hard-wired into the hardware, or can it be configured in software/firmare?

* If Table 3 and Table 4 were combined it would be clearer and easier to see how the proposed quantization schemes degrade accuracy relative to the baselines as well as other schemes.

* Section 5.1.2 - Please describe the method you used for quantization-aware training.

* Table 6: It would be helpful if this table somehow highlighted the one or tow variants of the proposed approach that the authors feel offers the most compelling overall tradeoff.  Is it SWIS-DS with #S=4?.  As it is, it's hard to tell what to take from Table 6.  Scatter plots of accuracy vs F/J and accuracy vs F/S would also be helpful.




**Paper Strengths:**

* The paper is well-written and easy to follow.
* The combination of software model evaluation and hardware simulation results is nice.  It is very helpful to be able to see the tradeoffs directly between hardware metrics like area and power and model accuracy.
* The information on model sensitivity would likely be useful to other researchers exploring compression methods.

**Paper Weaknesses:**

* The novelty in the compression scheme is minor.  Bit-masking approaches had been explored in previous works, so the main novelty is that the active bits need not be contiguous.

* The benefits of the proposed method are modest and vary from one model to another, so it is not clear whether they would be sufficient to justify the custom chip architecture and the work involved in specialized training routines (specifically for quantization-aware training variation).



**Poster (If Paper Is Rejected):**

1: Yes, ok for poster sesion to nurture work

**Reviewer Confidence:**

4: The reviewer is confident but not absolutely certain that the evaluation is correct

---

### Official Review · AnonReviewer3 · 2021-01-30

**Overall Merit Score:** 4

**Brief Summary:**

This paper presents a method to compress the weights of a neural network. The method exploits the fact that several of the bits in each weight are zero, stores them in a sparse representation. It then presents a bit-serial hardware architecture that leverages these compressed weights to improve execution time and energy.

**Detailed Comments:**

- This paper presents a method to compress the weights of a neural network. The method exploits the fact that several of the bits in each weight are zero, stores them in a sparse representation. It then presents a bit-serial hardware architecture that leverages these compressed weights to improve execution time and energy.

- Overall, this paper is well written, interesting and provides good tradeoff analysis. I liked the fact that instead of presenting one set of parameters that show benefits, the paper analyzes what block sizes, shift values, parallelism in PEs etc work well, and which ones don't. The results also comprehensively present accuracy, area, energy, and execution time results.

- It would be very useful to see a comparison with a non bit-serial architecture - a systolic array with a regular fixed point MAC and simple zero data gating.

- It would also be very useful to evaluate how this technique performs in conjunction with pruning algorithms, and whether once you use pruning, is the benefit significant compared to the above baseline with data gating.

- Finally, it would be helpful to compare with an architecture that allows more flexible dataflows, and evaluate whether or not the limitations that the weight-serial technique enforces on dataflow make it worse that the flexible non-serial architecture.


**Paper Strengths:**

- Overall, this paper is well written, interesting and provides good tradeoff analysis. I liked the fact that instead of presenting one set of parameters that show benefits, the paper analyzes what block sizes, shift values, parallelism in PEs etc work well, and which ones don't. The results also comprehensively present accuracy, area, energy, and execution time results.

**Paper Weaknesses:**

- It would be very useful to see a comparison with a non bit-serial architecture - a systolic array with a regular fixed point MAC and simple zero data gating.

- It would also be very useful to evaluate how this technique performs in conjunction with pruning algorithms, and whether once you use pruning, is the benefit significant compared to the above baseline with data gating.

- Finally, it would be helpful to compare with an architecture that allows more flexible dataflows, and evaluate whether or not the limitations that the weight-serial technique enforces on dataflow make it worse that the flexible non-serial architecture.

**Poster (If Paper Is Rejected):**

1: Yes, ok for poster sesion to nurture work

**Reviewer Confidence:**

4: The reviewer is confident but not absolutely certain that the evaluation is correct

---

### Official Review · AnonReviewer4 · 2021-01-31

**Overall Merit Score:** 3

**Brief Summary:**

This paper presents a quantization framework and its validation on bit-serial computational HW.

**Detailed Comments:**

see above

**Paper Strengths:**

- I liked the overall motivation and validation on post-synthesis HW. Overall, the framework is well demonstrated and efficient.
- quantization is applied in an interested way on weights. This results in a more compressed representation and efficient HW.
- Good weight compression is achieved at iso-accuracy. This is demonstrated on relevant systolic array architecture and post-synthesis evaluations are provided



**Paper Weaknesses:**

- even if this approach is not new, I found the overall evaluation valuable to TinyML conference

**Poster (If Paper Is Rejected):**

1: Yes, ok for poster sesion to nurture work

**Reviewer Confidence:**

4: The reviewer is confident but not absolutely certain that the evaluation is correct

---

### Decision · Program_Chairs · 2021-02-05

**Decision:**

Accept (Regular)

**Comment:**

Congratulations on your paper's acceptance!

Your paper has been accepted as a full-length regular paper.

Please read the reviews carefully and make sure the concerns are addressed in your final submission.

All accepted papers will be given a slot in the TinyML Summit schedule for an oral presentation on Friday, March 26, 2021.

Camera ready instructions will follow soon. All papers will be hosted on arXiv and published papers will have the following header stamp: “Published as a conference paper at TinyML Research Symposium 2021.” The paper will also be presented on the program website.